# Examining Protection Motivation and Network Externality Perspective Regarding the Continued Intention to Use M-Health Apps

**DOI:** 10.3390/ijerph18115684

**Published:** 2021-05-26

**Authors:** Yumei Luo, Guiping Wang, Yuwei Li, Qiongwei Ye

**Affiliations:** 1College of Business and Tourism Management, Yunnan University, Kunming 650091, China; luoyumei@ynu.edu.cn (Y.L.); Carolwgp@163.com (G.W.); liyuwei236@163.com (Y.L.); 2Business School, Yunnan University of Finance and Economics, Kunming 650221, China

**Keywords:** m-health apps, protection motivation theory, network externality, continued intention, perceived vulnerability, self-efficacy, response efficacy

## Abstract

M-health apps have developed rapidly and are widely accepted, but users’ continued intention to use m-health apps has not been fully explored. This study was designed to obtain a better understanding of users’ continued intention to use m-health apps. We developed a theoretical model by incorporating the protection motivation theory and network externalities and conducted an empirical study of a 368-respondent sample. The results showed that: (1) perceived vulnerability has a direct impact on users’ self-efficacy and response efficacy; (2) self-efficacy and response efficacy have a direct impact on users’ attitudes and continued intention; (3) network externalities affect users’ attitudes and continued intention, among which direct network externalities have an indirect impact on users’ continued intention through attitude; and (4) the impacts of self-efficacy, response efficacy, and indirect network externalities on continued intention are partially meditated by attitudes.

## 1. Introduction

M-health apps play an important role in the digitalization of nationwide healthcare services for better health outcomes due to the uniquity of smartphones in society [1,2]. In 2017, the number of available m-health apps was estimated to be approximately 300,000 and will grow by about 25% every year [3,4]. Although the potential of m-health apps is considerable, the current retention rate of actual m-health app users is comparatively low [5,6]. Due to the plethora of available m-health apps [3,4], there is fierce market competition [7]. The outbreak of COVID-19 in late 2019 further prompted a sharp increase in the demand for m-health apps [8]. This suggests a need to delve more deeply into m-health app users’ continued behaviors. Therefore, the present study pursued an exploration of the psychological mechanism that determines m-health app users’ continued intention to use m-health apps.

A robust body of m-health research has analyzed the factors influencing users’ intention to accept or not accept (e.g., [9,10,11,12]). Recently, some studies have focused on the continued behaviors of m-health app users [13,14]. These studies used some typical theoretical models, such as the post-adoptive model (PAM) and the expectation–confirmation model (ECM), and found that perceived usefulness, perceived ease of use, confirmation, and satisfaction were significantly associated with the continued intention to use health apps [13,14]. While these studies provide important insights into the factors influencing the continued use of m-health apps, studies that aim to understand other motivational factors, such as security motivation and network externalities motivation, remain limited.

M-health apps have a unique feature, that is, they provide health-related services [15]. Thus, besides being an emergent technology, users’ use of m-health may also be a health behavior triggered by the intention to avoid health threats and maintain security [16]. In addition, the social features of m-health apps facilitate communication and interaction between users, generating strong network externalities [17], which have also been ignored in studies on the continued usage of m-health apps.

To address this gap in our knowledge of the continued use of m-health apps, this study focuses on the impacts of two motivations (security and network externalities) on continued usage. The protection motivation theory (PMT) has been posited in order to study the effects of the security motivation on human behavior [18] and is widely used in health and environmental areas [19]. Network externalities are associated with users’ intention to adopt interactive information technologies [20,21]. However, in the context of continued usage of m-health apps, studies based on these theoretical perspective are seldom.

This study integrates the PMT and network externalities to provide insights into the relationship between PMT components (i.e., perceived vulnerability, self-efficacy, and response efficacy), direct and indirect network externalities, and continued intention by incorporating the mediating role of attitude to continued usage. This study enriches previous research on the continued use of m-health apps by considering the unique functional characteristics (i.e., security) of the chosen technology. The application of the PMT and network externalities in the m-health app context expands the theoretical scope of these two theories. We also hope to provide practical guidance to health service providers.

## 2. Literature Review

### 2.1. M-Health Apps

Mobile health refers to the provision of medical services to users through mobile devices [22] and has advantages over e-health in terms of convenience, mobility, and popularity due to its high degree of personalization and interactivity [17]. M-health apps can be divided into five types by function [23]. The first type comprises clinical and diagnostic apps. Through these apps, doctors can provide a diagnosis and treatment via pictures, videos, and calls for free or a certain fee. The second type comprises registration/appointment/guidance apps, which can help users reduce the time required for registration, treatment, and payment and hospitals optimize the resource allocation and service process, with the aim of improving the user’s experience during medical treatment. The third type comprises pharmaceutical service apps. The core of this type of app is to provide professional pharmaceutical counseling and counseling for minor ailments. The fourth type comprises health management (diet and data monitoring) apps. These platforms provide customized health plans and encourage users to participate in various activities and upload their own physical, health, diet, and sleep data to form user health files. The fifth type comprises vertical segment market apps. This type of app focuses on a specific market segment with social networks built among users based on similar characteristics, such as patients’ mutual aid and experience sharing.

“Reports on Chinese M-health Apps 2018” [24] showed that clinical and diagnostic apps have the highest user rate (41.5%). In February 2018, the top three apps regarding the number of monthly active users in China were Ping An Good Doctor, Micro Doctor, and Haodf, all of which are clinical and diagnostic apps. The 2019 iResearch data show that 65.9% of users use clinical and diagnostic apps when they face health issues. Clinical and diagnostic apps construct a communication channel with a high degree of interactivity among users and between doctors and patients [17]. Therefore, in this study, clinical and diagnostic apps were taken as the research objects to discuss continued usage behavior.

### 2.2. Protection Motivation Theory

The protection motivation theory (PMT), which originates from the health belief model [25], is used to study the impact of the security motivation on human behavior [18,19]. PMT discusses changes in health behaviors from the perspective of motivation factors [26] and has been widely used in research on health behavior change.

The PMT was proposed by Rogers [27] in 1975. Stainback and Rogers [28] integrated intrinsic rewards (e.g., bodily pleasure and satisfaction) and extrinsic rewards (e.g., social approval) into it and divided the PMT into three parts, namely sources of information, the cognitive mediation process, and coping modes, based on patterns of behavior formation. The cognitive mediation process is the core of this theory and includes threat appraisal and coping appraisal. The cognitive mediation process is initiated by sources of information and ultimately comes down to coping modes. Sources of information include personal factors and external environmental factors, while coping modes include adaptive responses (e.g., unhealthy behavior changes) and maladaptive responses (e.g., unhealthy behavior continuance).

Among them, threat appraisal is an assessment of unhealthy behaviors or diseases [29], including intrinsic rewards, extrinsic rewards, perceived severity, and perceived vulnerability. Intrinsic rewards refer to the self-satisfaction or inner positive feelings experienced by individuals when they participate in a specific activity or behavior, while extrinsic rewards refer to factors coming from peers, the family, or other social groups that can strengthen an individual’s behavior [26]. Perceived severity refers to an individual’s judgment on the hazard that certain behavior represents to his/her physical and mental health. Perceived vulnerability refers to one’s subjective judgments on the possibility of certain diseases [30]. Therefore, threat appraisal is an integration of the intrinsic rewards, extrinsic rewards, perceived severity, and perceived vulnerability of behavior [26]. Coping appraisal is an evaluation of an individual’s ability to cope with and avoid risks, which includes the individual’s self-efficacy and response efficacy. Self-efficacy refers to one’s perceived ability to take protective behaviors (e.g., using m-health apps to find health-related information or services), and response efficacy refers to one’s perception of the effectiveness of certain protective behaviors (e.g., getting timely feedback from m-health apps) [30].

Therefore, the PMT comprehensively analyzes the causes and processes of behaviors from external sources of information and internal cognitive characteristics [29]. It takes threat appraisal and coping appraisal as processes and predicts whether there will be protective motivations and behaviors according to the results of comprehensive evaluations [18] and then explains the process of a change in behavior.

The PMT has been widely applied in the study of behavioral changes in various public health settings, such as smoking, education about chronic diseases, and AIDS prevention, to explain, predict, and intervene into health behaviors [26,29]. Studies on the relationship between the PMT, attitude, and intention found that threat appraisal and coping appraisal significantly affect an individual’s attitude and behavior [19,31,32] and intention to use [18]. Some studies also found that the PMT first influences attitude, then intention [16], and finally user behavior [30]. Therefore, based on the PMT, this study believed that threat appraisal and coping appraisal are associated with the attitude toward and intention to continued use of m-health apps.

Intrinsic rewards and extrinsic rewards in threat appraisal are factors resisting healthy behaviors and are used to explain an individual’s refusal to engage in healthy behaviors [26]. This study is targeted at the users of clinical and diagnostic apps who show healthy behaviors and do not resist healthy behaviors. Perceived severity in threat appraisal refers to individuals’ judgments on unhealthy behaviors or the severity of diseases. Users of clinical and diagnostic apps mainly care about the possibility of whether they have a certain kind of disease, i.e., perceived vulnerability, and care less about disease severity, i.e., perceived severity. Therefore, this study adopts perceived vulnerability in threat appraisal and self-efficacy and response efficacy in coping appraisal. In addition, in the context of m-health apps, using m-health apps is regarded as a kind of protective behavior. Thus, self-efficacy refers to one’s judgment on his/her ability to obtain benefits from continued use of clinical and diagnostic apps, which is similar to e-health literacy. Response efficacy refers to an individual’s perception of the effectiveness of continuously adopting protective behaviors, such as using clinical and diagnostic apps.

### 2.3. Network Externalities

In recent years, network externalities, as a key predictor of users’ intention to adopt interactive information technology [20], have been widely used to examine user behaviors in information systems [33]. Therefore, network externalities are taken as a foundation of the model in this study and to provide theoretical support to explore the intention to continue to use interactive m-health apps.

Network effects refer to the effects or values that users derive from other users who use similar or compatible products [34]. Network effects arise when the utility that consumers derive from the consumption of a product or service depends on the number of other users of the same product or service or the availability of complementary products or services that generate additional value for users of the original product or service [20]. Factors that drive network effects, such as network size and the utility of complementary products or services, are network externalities [35].

Network externalities are divided into direct network externalities and indirect network externalities [18,20,36]. Direct network externalities are related to the number of network users. For example, the number of players on an online gaming website. As new players enter, existing users will have more opportunities to trade, communicate, and play, thus generating network effects. Indirect network externalities refer to the additional benefits associated with user diffusion [36], such as the complementary services provided by third parties, and are not directly derived from the number of network users. In recent years, network externalities have received attention in information systems research [36] and have been applied to examine the behavior of information system users [33]. Lin and Bhattacherjee [20], drawing on theories of traditional usage motivations and network externalities, studied the factors influencing the continued use of instant messaging (IM). The research showed that network externalities (namely the referent network size and perceived complementarity) have a direct impact on the continued intention to use interactive information technology. Zhou [36], by integrating the external motivation theory and the Flow theory, found that referent network size and perceived complementarity, two factors driving network externalities, have a significant impact on flow and further affect users’ satisfaction with and continued usage of mobile social network services (SNS). Zhou, Li, and Liu [33] synthesized network externalities and switching costs and proposed a model of the continued usage of mobile instant messaging services. They found that the perceived complementarity is a major factor influencing perceived enjoyment, and referent network size has a significant impact on perceived usefulness. Cheng [37] integrated subjective norms, network externalities, task–technology fit (TTF), and the expectation–confirmation model (ECM) to explain nurses’ continued intention to use an e-learning system in medical institutions. The results indicate that the user network profoundly affects nurses’ intention to continue usage.

Based on the above-described literature, network externalities have been widely used in research on different interactive information technologies, such as IM, mobile SNS, e-learning, and wechat. However, network externalities have rarely been applied in m-health. Therefore, this study introduced network externalities to enhance our understanding of the continued intention to use m-health.

## 3. Research Model and Hypotheses

This study explores the psychological mechanism of users’ continued intention to use m-health apps from the perspectives of the protection motivation and network externalities as shown in Figure 1. The two dashed blocks stand for two theoretical perspectives, namely protective motivation and network externalities; the blocks with a solid line border stand for constructs in the research model; and the lines with arrows stand for the hypothetical relationship between the constructs. We first established the theoretical linkage between perceived vulnerability and self-efficacy (H1) and between perceived vulnerability and response efficacy (H2). We then articulated the connection between protective motivation and attitude (H3a and H3b) and continued intention (H4a and H4b), respectively. From a network externality perspective, we explain the relationship between direct network externalities and attitude (H5a) and continued intention (H5b) as well as the association between indirection network externalities and attitude (H6a) and continued intention (H6b). Finally, we elucidate the direct role (H7) and mediator role (H8a–H8d) of attitude.

### 3.1. Hypotheses Based on PMT

According to the PMT, perceived vulnerability refers to an individual’s perception of the probability that a certain threat will occur [29]. Self-efficacy refers to one’s perception of their ability to adopt protective behaviors; in other words, the beliefs, judgments, and subjective self-perceptions one has for completing an action before performing it [30]. Response efficacy refers to the perceived effectiveness of certain healthy behaviors [29]. In this study, perceived vulnerability refers to an individual’s perception of the possibility of experiencing the serious disease. Self-efficacy refers to one’s judgment on his/her ability to obtain benefits from continued use of clinical and diagnostic apps, such as obtaining health-related information, guidance, and services to prevent or deal with health threats, which is similar to e-health literacy. Response efficacy refers to the perception of the effectiveness of the continuous adoption of protective behaviors, specifically the use of clinical and diagnostic apps.

Based on the PMT, information about health threats obtained from the environment and other people can facilitate an individual’s judgment on this information [26]; in other words, to stimulate the threat appraisal and the coping appraisal in the cognitive mediation process [38]. During threat appraisal, self-efficacy and response efficacy are initiated by one’s perceived vulnerability to threats and fears [19]. Perceived vulnerability is positioned as a direct prerequisite for self-efficacy and response efficacy [20]. A coping appraisal is only performed when threats and fears are related and potentially harmful (when the perceived vulnerability is high) [20]. Therefore, in the context of m-health apps, m-health apps could provide a vast amount of health-related information and clinical and diagnostic services. Using m-health apps is regarded as a kind of protective behavior to deal with threats. When the perceived vulnerability of certain diseases increases, an assessment of protective behavior (i.e., using m-health apps) will be motivated. The individual will evaluate his/her ability to use such apps and also assess the function of such apps. Therefore, we propose the following hypotheses:

**Hypothesis** **1.**
*Perceived vulnerability has a significant positive effect on users’ self-efficacy of using m-health apps.*


**Hypothesis** **2.**
*Perceived vulnerability has a significant positive effect on users’ response efficacy of using m-health apps.*


Attitude refers to the positive or negative emotion that individuals have towards certain behaviors [39]. In this paper, the attitude toward continued use of m-health apps is defined as an individual’s positive or negative emotion towards m-health apps. Attitude will be jointly determined by perceived self-efficacy, response efficacy, and network externalities.

In a study on computer security behavior, Anderson and Agarwal [19] found that users’ perceived threats and self-efficacy have a positive impact on users’ attitudes towards security behaviors. Some researchers also found that the threat appraisal and the coping appraisal in the PMT affect attitudes first, and then influence user behavior [40]. Yoon and Kim [32] found that perceived severity and response efficacy in the PMT are positively related to employee attitude, which in turn has a significant positive impact on the employee’ intention to practice computer security behavior.

Based on the PMT, individuals with higher perceived competence in protective behaviors are more confident in benefiting from these behaviors and will have stronger behavioral motivations [30]. Perceived competence in using information technology, i.e., self-efficacy, is a key influencing factor in research on user behavior [41,42]. When faced with security threats, individuals will use certain protective behaviors to deal with threats. For example, they will use m-health apps to find health-related information and services and to evaluate these threats. If they have stronger self-efficacy to use m-health apps, they are more confident that they will benefit from m-health apps and be able to solve health issues. The confidence in using m-health apps as a precautionary measure will motivate them to believe in the necessity of taking precautions to deal with these threats.

**Hypothesis** **3a.**
*The self-efficacy of using m-health apps has a significant positive effect on users’ attitudes towards using m-health apps.*


Similarly, when individuals have a higher perceived response efficacy to m-health apps, they are less likely to omit to perform threat prevention actions. Security studies suggest that individuals with a higher perceived response efficacy are less likely to omit to perform security actions in the workplace [40] and are more likely to use a firewall in their home wireless network [43]. Therefore, if an individual believes that using m-health apps is helpful, he/she is more likely to believe that their behavior can bring about change.

**Hypothesis** **3b.**
*The response efficacy to m-health apps has a significant positive effect on users’ attitudes towards using m-health apps.*


At the same time, the self-efficacy and response efficacy of using m-health apps will directly affect users’ continued intention to use m-health apps. The PMT indicates that, in most cases, perception will directly affect the intention of behavior [27,38]. Self-efficacy and response efficacy increase the possibility of an adaptive response, and people believe that they will benefit from adopting certain behaviors [26]. When faced with threats, individuals with higher competence in dealing with threats are more likely to change their behaviors [19,30]. The direct effect of self-efficacy and response efficacy on behavioral awareness has also been recognized by many studies [18,30]. Therefore,

**Hypothesis** **4a.**
*The self-efficacy of using m-health apps has a significant positive effect on users’ continued intention to use m-health apps.*


**Hypothesis** **4b.**
*The response efficacy of using m-health apps has a significant positive effect on users’ continued intention to use m-health apps.*


### 3.2. Hypotheses Based on Network Externalities

#### 3.2.1. Direct Network Externalities

As mentioned above, direct network externalities are related to the number of network users. Interactive information technology users usually decide their usage behavior based on the number of friends, colleagues, or others within their social network who are using this technology [20]. In the context of mobile healthcare, direct network externalities refer to the perceived number of users using m-health apps. Upon first use, users will spend more time getting to know the functions and features of m-health apps as they will be encouraged by the large user group and the intention of group convergence, even if they know little about the app [44]. At the same time, as the user number increases, users can obtain more health information and better services, which come along with more sharing, communication, and mutual help among users to deliver more network benefits. In this way, users will have a more positive attitude towards m-health apps.

**Hypothesis** **5a.**
*Direct network externalities have a significant positive effect on the attitude towards the continued intention to use m-health apps.*


In addition, potential users will be affected by the number of existing users of certain m-health apps when deciding to adopt and continue to use m-health apps [45]. In addition, Wang et al. [46] found that models that integrate network externalities have stronger explanatory power regarding users’ intention to use information technology. Users perceive more network externalities when the network size is large, which leads to a stronger intention and a higher intensity of continued use. In the context of m-health apps, the perceived number and activity of users will further promote a sense of collective action and a strong perception of safety, thereby promoting users’ willingness to continue using the apps.

**Hypothesis** **5b.**
*Direct network externalities have a significant positive effect on users’ continued intention to use m-health apps.*


#### 3.2.2. Indirect Network Externalities

Indirect network externalities come from the available complementary products or services; in other words, the value-added functions and services that provide additional network benefits to users [20,36]. In the context of mobile healthcare, indirect network externalities are defined as the complementary and value-added functions or services provided by m-health apps (such as health management tools, health discussion groups, online shopping, outfits, and skincare). A study on the continued usage of mobile IM found that as the number of users continued to increase, service providers of mobile IM frequently updated their products and services with more functions, such as photo sharing, music, video, and games [33]. These complementary functions and services promote users’ attitudes towards and continued intention to use mobile IM. These complementary services and functions also facilitate users to not only communicate with more users but also to acquire complementary and value-added services through a single platform. This will have a positive impact on users’ attitudes toward usage [47] and improve users’ satisfaction [36]. Prior research also identified a positive correlation between indirect network externalities and continued intention of use [37]. Therefore, the following hypotheses are proposed:

**Hypothesis** **6a.**
*Indirect network externalities have a significant positive effect on users’ attitudes towards the continued usage of m-health apps.*


**Hypothesis** **6b.**
*Indirect network externalities have a significant positive effect on users’ continued intention to use m-health apps.*


### 3.3. Attitudes

The relationship between attitudes and behavioral intentions has been theorized about and extensively tested in a robust body of literature from multiple disciplines [48,49]. Attitudes toward specific actions influence intentions to perform the actions because individuals seek a cognitive consonance between feelings and actions [50]. Therefore, the following hypothesis is proposed:

**Hypothesis** **7.**
*Users’ attitude towards m-health apps has a significant positive effect on their continued intention to use m-health apps.*


At the same time, a robust body of literature also supports the mediating role of attitude. It was found that security threats, perceived response efficacy, and self-efficacy affect an individual’s attitude first and then their behavior [19]. When the perceived vulnerability increases, people’s attitudes will change and their intention to change their behavior will also become stronger [40]. Yoon and Kim [32] found that perceived vulnerability and response efficacy are positively correlated with attitude, and attitude has a significant positive impact on the intention to perform computer security actions. Zhang et al. [51] also verified the mediating role of attitude. In a study on the effects of network externalities on users’ continued intention to use, Li et al. [52] also found that attitude has a mediating effect on users’ continued intention to use. Therefore, the following hypotheses are proposed:

**Hypothesis** **8.**
*Users’ attitude towards m-health apps has a significant mediating effect, specifically:*


**Hypothesis** **8a.**
*Users’ attitude towards m-health apps mediates the effect of self-efficacy on the continued intention to use.*


**Hypothesis** **8b.**
*Users’ attitude towards m-health apps mediates the effect of response efficacy on the continued intention to use.*


**Hypothesis** **8c.**
*Users’ attitude towards m-health apps mediates the effect of direct network externalities on the continued intention to use.*


**Hypothesis** **8d.**
*Users’ attitude towards m-health apps mediates the effect of indirect network externalities on the continued intention to use.*


## 4. Methodology

### 4.1. Data Collection

We recruited participants through Sojump (https://www.wjx.cn/ (accessed on 29 May 2020)), a professional questionnaire distribution platform in China. Participants were randomly recruited from across the country. In order to identify target participants, we first asked the question: “Have you ever used clinical and diagnostic m-health apps?” If he/she answered “no” to the question, the questionnaire jumped to the end. Only those people who used clinical and diagnostic m-health apps were included as respondents in this study. Respondents who completed and submitted the questionnaire were rewarded with 2 RMB on average. The survey started on 30 May 2020 and lasted for half a month. A total of 400 questionnaires were collected. Invalid questionnaires that were incomplete or had obvious errors, such as all 1 or all 7, were excluded. A total of 368 valid questionnaires were obtained.

Based on the statistical analysis, the descriptive statistics of the samples are shown in Table 1, with 62% males and 38% females. 50.5% of participants were aged 26–35 and 74.8% of participants had a bachelor’s degree or above. In terms of health conditions, 44.6% of participants were in a sub-health condition.

### 4.2. Measures

Seven constructs were measured in this study: behavioral intention (BI), attitude (ATTI), direct network externalities (DNE), indirect network externalities (INE), self-efficacy (SE), response efficacy (RE), and perceived vulnerability (PV) (See Table 2). The constructs were multi-item scales drawn from previously validated measures and were adapted to relate specifically to the context of the use of m-health apps. Behavioral intention to continue using m-health apps (CI) and attitude toward m-health apps (ATTI) were measured using three-item scales adapted from Venkatesh and Goyal [53] that have been used extensively in prior research. DNE and INE were measured using three-item scales adapted from Lin and Bhattacherjee [20]. SE, PV, and RE were measured using scales adapted from Johnston and Warkentin [18]. All items were assessed via a seven-point Likert scale, with “1” representing “strongly disagree” and “7” representing “strongly agree”.

We also measured a few control variables, including age, education, gender, and health condition.

Two senior information systems (IS) professors followed back-translation procedures to convert the survey instrument from English to Chinese [54]. We invited several graduate students familiar with research methods to carefully examine the translated questionnaire and made further amendments based on their feedback. We then conducted a pilot test using 30 users of m-health apps. The pilot test data were analyzed using exploratory factor analysis with Varimax rotation to examine whether the scale items loaded up on their hypothesized scales as expected, yielding a seven-factor structure that explained 85.4% of the variance in the data. The primary factor loading was 0.4 or above. This process of instrument refinement led to considerable improvement in the construct validity and reliability of our measurement scales. The pilot test responses were excluded from our larger hypotheses-testing survey.

The final version of the survey instrument for formal data collection includes three sections. In the first section, we briefly introduce the background of and motivation for our study and the concepts of different m-health apps and obtain the consent of the participant. In the second section, we ask participants to faithfully evaluate their perceived vulnerability, self-efficacy, response efficacy, direct network externalities, indirect network externalities, attitude, and intention to continue using m-health apps. In the final section, participants are asked to fill in demographic information, including their gender, age, educational level, and health condition.

### 4.3. Results 

SmartPLS 3.0 [55] was applied to analyze the data. Within the context of the theoretical mediated model, PLS’s ability to assess the measurement model is superior to multiple regression. Our model consists of measurement models and structural models. Before testing the structural models, we assessed the reliability and validity of the measurement models.

#### 4.3.1. Measurement Model

Table 3 shows the results from the measurement model, which include information about reliability, convergent validity, and discriminant validity. First, construct reliability was supported because all the Cronbach’s alpha and composite reliability values satisfied the criteria of 0.70 [56]. The convergent validity of all constructs was also supported because all items displayed a factor loading higher than the reference value of 0.70, ranging from 0.773 to 0.936 (see the last row of Table 3), and all AVEs exceeded the threshold of 0.50, ranging from 0.688 to 0.851. Finally, according to Fornell and Larcker [57], for a construct to demonstrate discriminant validity, the square root of the AVE of any construct must be higher than its correlation with other constructs.

#### 4.3.2. Structural Model

The results from our structural model test (Figure 2) support the hypothesized relationships. Through using a bootstrapping resampling procedure, the analysis produced estimates of both the path coefficient as well as the explained variance in SE, RE, ATTI, and CI. Of the hypotheses about direct effects, all but the influence of DNE on CI were found to be significant. Of the four hypotheses about indirect effects through ATTI, all were found to be significant. The explained variance for the model was also reasonable. Overall, we concluded that the model has received good support.

As indicated in Figure 2, the model explains approximately 19 percent, 18.6 percent, 43.7 percent, and 62.2% percent of the variance. The highest explanatory power of 62.2 percent was found for the path for INE, ATTI, SE, and RE leading to continued intention. 

The results of the structural model analysis also confirm the positive relationships between PV, SE, and RE. Specifically, PV has significant positive effects on SE (β = 0.403 ***, *p* = 0.000) and RE (β = 0.329 ***, *p* = 0.000), supporting H1 and H2, respectively. In addition, both SE and RE were positively related to ATTI (β = 0.178 **, *p* = 0.003 for SE; β = 0.230 ***, *p* = 0.000 for RE), supporting H3a and H3b, and CI (β = 0.208 **, *p* = 0.006 for SE; β = 0.221 ***, *p* = 0.000 for RE), supporting H4a and H4b.

Regarding the effect of network externalities, DNE was positively related to ATTI (β = 0.104, *p* = 0.025) but exhibited no significant influence on CI (β = 0.031NS, *p* = 0.516). Therefore, H5a was supported and H5b was not. INE had positive effects on ATTI (β = 0.258 ***, *p* = 0.000) and CI (β = 0.207 **, *p* = 0.001), supporting H6a and H6b. In addition, ATTI had a positive influence on CI (β = 0.271 ***, *p* = 0.000), supporting H7.

Our hypotheses posited that the effects of SE, RE, DNE, and INE on CI are mediated by ATTI. We followed the mediation tests suggested by Baron and Harris [58] and used the bootstrapping method to assess the significance of mediating effects. As shown in Table 4, the bootstrap results of the indirect effect of SE, RE, DNE, and INE on CI through ATTI revealed bias-corrected 95% confidence intervals that did not involve zero (β = 0.048, BCCI (0.008, 0.089) for self-efficacy to CI; β = 0.023, BCCI (0.017, 0.108) for RE to CI; β = 0.028, BCCI (0.002, 0.054) for DNE to CI; and β = 0.070, BCCI (0.028, 0.111) for INE to CI). Considering their direct effects, these results suggest that ATTI partially mediated the effect of SE, RE, and INE on CI and completely mediated the effect of DNE on CI, therefore supporting H8a–H8d.

## 5. Discussion

Based on the protection motivation theory and network externalities, this study examined the impact of perceived vulnerability, self-efficacy, response efficacy, and direct and indirect network externalities on users’ attitudes toward and continued intention to use m-health apps. The mediating effect of attitude was further verified. Overall, the empirical results support most of our hypotheses, except for one path relationship (H5b). Our specific findings are as follows.

First, this study found that the perceived vulnerability to diseases has a positive impact on self-efficacy and response efficacy in the cognitive mediation process (H1 and H2). This indicates that with a higher perceived vulnerability, individuals will be more likely to evaluate the self-efficacy of taking protective measures and the effectiveness of the measures, which will drive the assessment of self-efficacy and response efficacy. Individuals with a stronger perceived ability to perform protection actions are more confident in benefiting from these protection actions. Additionally, it is more likely that they will adopt a positive attitude towards protective behaviors to change their situations (H3a and H3b), stimulating and complying with healthy behavior or limiting unhealthy behavior. In this context, individuals will continue to use m-health apps (H4a and H4b).

Second, for m-health apps, the types of technologies or services with interactive functions, i.e., network externalities, have an important impact on users’ attitudes and behavior. Previous studies have tested this hypothesis in different contexts [20]. This study provides further support to previous conclusions and found that direct and indirect network externalities have significant positive effects on users’ attitudes (H5a and H6a) but do not have consistent effects on continued use behavior. In other words, the hypothesis regarding a direct effect of direct network externalities on continued intention is not supported (H5b). The effect of direct network externalities on the continued intention to use is mediated by attitude (H8c). On the contrary, the direct effect of indirect network externalities is supported (H6b). All these results indicate that both the network size and the complementary functions and services provided by the network will exert network externalities on the users of m-health apps, thus promoting a direct effect on attitude and direct or indirect effects on the continued intention to use.

Third, the study further verified the significant effect of attitude on behavior (H7) and the mediating effect of attitude on the relationship between these psychological mechanisms and continued behavior (H8a–H8d). Specifically, we found that attitude partially mediates the effect of self-efficacy, response efficacy, and indirect network externalities on the continued intention to use, while attitude completely mediated the impact of direct network externalities. Although extensive support has been obtained for the effect of attitude on behavior in different contexts, we found that attitude only partially mediates the effect of psychological mechanisms on users’ continued use behavior. On the one hand, it shows that attitude is not the only direct determinant of continued intention. On the other hand, it also indicates that, in the context of m-health apps, there may be other factors mediating the impact of psychological mechanisms on users’ continued intention to use. For example, Zhou [36] found that satisfaction and flow mediate the impact of network externalities on users’ continued intention to use mobile SNS. More opportunities for future research exist.

Finally, we analyzed demographic characteristics. The results show that users’ psychological mechanisms were mainly related to age and education level, but no gender and health condition. Specifically, the older the user, the stronger their self-efficacy and response efficacy, and the higher the level of education, the higher the response efficacy and continued intention to use m-health apps. This result is interesting because it is inconsistent with that in the general technology literature (i.e., men and youth are more positive about new technologies). The reason for this difference is that m-health is both a new technology and a health-related product [16]. Thus, when people make decisions on whether to continue to use it, they will also take their health-related factors into consideration. For instance, with an increase in age or an improvement in education level, users will have more health-related knowledge and experience. Thus, they will have more confidence when evaluating health-related information and services, and they will be more willing to use m-health apps.

### 5.1. Theoretical Implications

From a theoretical perspective, this study integrates the protection motivation theory and network externalities to test users’ continued intention to use m-health apps. As mentioned above, the existing research mainly discusses users’ behavior regarding m-health apps based on the technology acceptance model (TAM). Research on the continued usage of m-health apps is rare. Therefore, this study extended the research on continued use behavior to the scenario of m-health apps.

Second, considering that m-health apps are technology-based products and services related to health and diseases, their technology-specific functions and scenarios are different from other types of apps. It is necessary to explore their influencing factors considering their specific functions and scenario characteristics. Therefore, this study integrated the protection motivation theory to explore the influence of perceived vulnerability, self-efficacy, and response efficacy on user behavior. This not only enriches the research on continued use behavior but also further verifies the applicability of the protection motivation theory in the context of m-health apps.

Third, as an interactive form of information technology, m-health apps have strong network externalities. This study further verified the impact of network externalities on users’ continued intention to use m-health apps and found that the mechanisms of direct network externalities and indirect network externalities are different. These findings also further deepen our understanding of network externalities.

Finally, attitude has a widely accepted effect on behavior. However, this study found that attitude partially mediates the impact of related factors in the PTM and network externalities on users’ continuous use behavior, which indicates that attitude is not the only important factor that determines behavior. In health-related behavior, there may be other important factors that can be further discussed in the future.

### 5.2. Practical Implications

From a managerial perspective, our results imply that service providers need to be concerned with both the PTM and network externalities in order to facilitate users’ continued use. First, the results of this study support the use of fear-inducing arguments as an effective way to influence end-user intentions to continue to use m-health apps. Messages warning of new health threats will inspire users to accept the message and take appropriate action to reduce the threat. Thus, to effectively wield fear as a motivator, service providers must devise a strategy in which end users are exposed to appeals to fear to thus motivate users to actively evaluate and adopt protective behaviors.

Second, increasing users’ self-efficacy and response efficacy are effective ways of improving users’ attitude and continued intention to use m-health apps. Service providers should be fully aware of this fact and adopt effective ways to increase users’ self-efficacy and response efficacy. For example, service providers of m-health apps should strengthen the promotion of healthcare information and provide users with more health-related knowledge or convenient and understandable training and guidance (e.g., a vivid and interesting cartoon guide) to increase users’ self-efficacy. Meanwhile, from a system function perspective, improving the responsiveness of the system will help to promote users’ perceived response efficacy.

Third, attaching more importance to network externalities, including direct and indirect network externalities, may help to attract m-health app users. Service providers should be cognizant of and consider the role of network effects in their rollout plans. For instance, they should establish interactive functions that are easy to communicate and share, strengthen mutual communication between users, strengthen the communication between doctors and patients and patients and patients, and continuously expand the user network. Then, a diverse range of complementary value-added services, such as drug purchase platforms, training, and insurance, should be provided and fully integrated into medical and healthcare services to provide a one-stop service. When users perceive a positive network externality, they may appreciate the great utility of m-health apps and expect a compelling experience.

### 5.3. Limitations and Future Research

This research has the following limitations. First, our findings may not apply in other countries, as the survey was conducted in China. Cross-cultural research needs to be performed in the future. Second, our model only considers psychological mechanisms as variables; other variables, such as interface ergonomics and app design, may also affect people’s use of m-health apps. Future studies can be based on this research and add other influencing factors to complement this study. Third, we conducted a cross-sectional study. Thus, longitudinal research may provide more insights into user behavior. In addition, future research could monitor users’ actual behavior and use these objective data to conduct a data analysis. 

## Figures and Tables

**Figure 1 ijerph-18-05684-f001:**
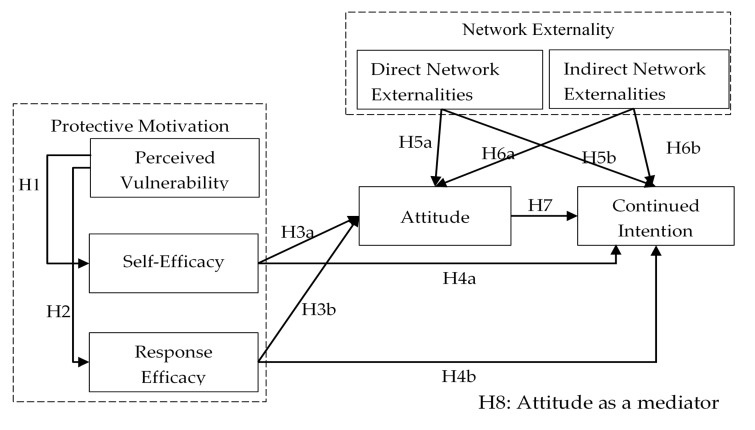
Research model and hypotheses.

**Figure 2 ijerph-18-05684-f002:**
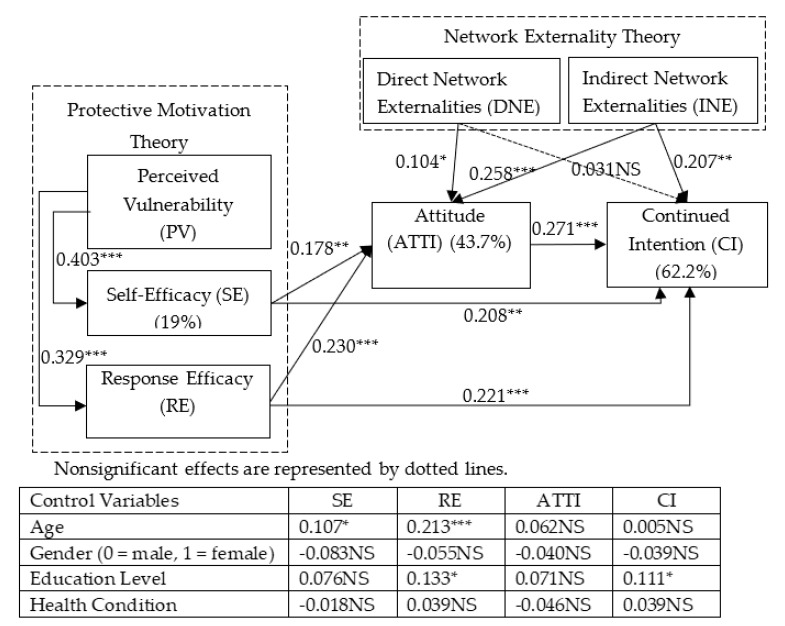
Results of the PLS structural model analysis. *** *p* < 0.001; ** *p* < 0.01; * *p* < 0.05; NS, nonsignificant.

**Table 1 ijerph-18-05684-t001:** Sample Demographics (*n* = 368).

Characteristics	Category	Frequency	Percentage (%)
Age	18–25	75	20.4
	26–35	186	50.5
	36–45	70	19.0
	46–65	37	10.1
Education	High school or below	38	10.4
	Junior college	62	16.8
	Bachelor’s degree	202	54.9
	Master’s degree or above	66	17.9
Gender	Female	140	38.1
	Male	228	61.9
Health condition	Healthy	156	42.4
	Sub-health	164	44.6
	Underlying minor illness	32	8.7
	Underlying chronic disease	16	4.3

**Table 2 ijerph-18-05684-t002:** Constructs, items, and references.

Constructs	Items	References
Self-Efficacy	SE1. I know what kind of health-related information is provided on this m-health app. SE2. I have the competence to assess the correctness of the health-related information provided on this m-health app. SE3. I can use this m-health app to make health-related decisions.	[18]
Perceived Vulnerability	PV1. I think I am facing the threat of serious disease. PV2. I think I am facing the probability of suffering from a serious disease in the future. PV3. I will probably suffer from a disease.	
Response Efficacy	RE1. This m-health app can notify users of the starting and ending time of healthcare services in time. RE2. This m-health app can send in-time feedback to me. RE3. This m-health app can provide instructions if I have some problems.	
Direct Network Externalities	DNE1. Most of my friends use this m-health app. DNE2. The majority of my colleagues use this m-health app. DNE3. The majority of the people I know use this m-health app.	[20]
Indirect Network Externalities	INE1. This m-health app provides many complementary services (e.g., health management tools and discussion groups). INE2. This m-health app provides many other application services. INE3. This m-health app provides some value-added services (online shopping, outfits, skincare, weight reduction and body shaping, medical cosmetology, etc.)	
Attitude	ATTI1. Using this m-health app is a good idea. ATTI2. Using this m-health app makes seeing a doctor easier and more enjoyable. ATTI3. I like to use this m-health app.	[53]
Continued Intention	CI1. I intend to continue using this m-health app in the future. CI2. I will continue to use this m-health app. CI3. I will recommend this m-health app to others.	

**Table 3 ijerph-18-05684-t003:** Descriptives, correlations, and measurement model statistics.

Constructs	PV	SE	RE	DNE	INE	ATTI	CI
Perceived Vulnerability (PV)	**0.875**						
Self-Efficacy (SE)	0.414	**0.840**					
Response Efficacy (RE)	0.362	0.670	**0.848**				
Direct Network Externalities (DNE)	0.276	0.514	0.529	**0.923**			
Indirect Network Externalities (INE)	0.333	0.604	0.606	0.478	**0.83**		
Attitude (ATTI)	0.266	0.550	0.574	0.447	0.563	**0.833**	
Continued Intention (CI)	0.293	0.645	0.664	0.470	0.642	0.651	**0.894**
Mean (SD)	3.894 (1.384)	4.913 (1.038)	5.158 (0.983)	4.530 (1.263)	4.997 (1.005)	5.495 (1.012)	5.418 (1.051)
Composite reliability	0.908	0.878	0.884	0.945	0.869	0.872	0.922
Cronbach’s alpha	0.851	0.791	0.804	0.912	0.775	0.779	0.874
CFA Item Loadings ^	0.814–0.917	0.812–0.879	0.837–0.855	0.915–0.936	0.773–0.853	0.808–0.871	0.877–0.910

Notes: The bold diagonal elements represent the square root of the average variance extracted (AVE). ^: The CFA loadings reflect the range of loadings (lowest loading to highest loading) that the items of each scale have on their latent construct.

**Table 4 ijerph-18-05684-t004:** Results of indirect effects.

Independent Variables	Mediator Variable	Dependent Variable	Indirect Effect Coefficients	95% Bias-Corrected Confidence Intervals	Hypotheses
SE	ATTI	CI	0.048 (0.020)	(0.008, 0.089)	H8a (√)
RE			0.062 (0.023)	(0.017, 0.108)	H8b (√)
DNE			0.028 (0.013)	(0.002, 0.054)	H8c (√)
INE			0.070 (0.021)	(0.028, 0.111)	H8d (√)

Notes: Standardized path coefficients with standard errors in parenthesis. SE, self-efficacy; RE, response efficacy; DNE, direct network externalities; INE, indirect network externalities; ATTI, attitude; CI, continued intention.

## Data Availability

The data presented in this study are available on request from the corresponding author.

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
