# Peer review of "Examining Protection Motivation and Network Externality Perspective Regarding the Continued Intention to Use M-Health Apps"

_ijerph, 2021, doi:10.3390/ijerph18115684_

Round 1

Reviewer 1 Report

The abstract needs to be reworked, it is not clear what is the paper about
Lines 76 - Pharmaceutical e-commerce services are normally not considered mHealth
Inclusion/criteria for the respondents are not described, it is not clear how the respondents were recruited, there is no description of the cohort
Table titles are above the table but should be bellow
Some abbreviations are not defined – like IS on line 366
The questionnaire should be better described
The whole experiment is not clearly described. Although there are some practical implications, they are still very general and abstract and there is no justification for these implication regarding the results from this research.

Author Response

In the document  we provide point-by-point responses on how we have addressed each comment from the reviewer. We sincerely hope that the changes in this revision meet the review panel’s expectations. Thank you again.

Reviewer 2 Report

This is an interesting and well-written manuscript regarding factors that affect m-health app users' intentions to continue using these apps. The work conducted employs the theories of protection motivation and network externalities to explore their impact on continuance intention, which is claimed by the authors to be the innovative part of this work.

Overall, the paper is well-written and the work done is thorough.

Self-efficacy as used traditionally in technology acceptance models, refers to one's convictions about his or her abilities to mobilize motivation, cognitive resources and courses of action needed to successfully execute a specific task within a given context (e.g., computer self-efficacy is defined as an individual judgement of one’s capability to use a computer). In the context of the PMT, self-efficacy  refers  to one’s perceived ability to take protective  behaviors. This may cause a confusion to readers familiar with the TAM. Perhaps the authors could clarify how these two views on self-efficacy are different.

In p. 6 it is mentioned that "stronger self-efficacy and response efficacy indicate higher confidence of users in benefiting from m-health apps, in other words,users are more confident to solve health issues and with stronger perceived usefulness". In my understanding the PMT does not refer to using particular technologies. It refers to applying protective behavior. In the current Covid-19 context, such a behavior could be to wear a mask and wash one's hands often - not using technology at all. In brief, it is not clear how from the PMT theory the authors jump to conclusions about m-health apps. This becomes clear later through the questionnaires, however authors could perhaps further elaborate on this in p.6 and follow a clear path in their line of thinking in order to alleviate any misconceptions. Generally, it would be good to provide early examples of what the authors have assessed through the questionnaires when the terms are introduced.

An important concern refers to the lack of connection with TAM and concepts such as perceived ease of use and perceived usefulness. This gap is attempted to be bridged at the last part of the paper discussing practical implications. However, at this point, it is not clear how the authors arrive to the conclusion that "developers and operators of m-health apps should improve the medical and healthcare functionsto meet users’ needs and ensure the accuracy of information and services, which will enhance  users' perception  of response  efficacy." Is this based on data or observations from the current research? The same holds for the third practical implication mentioned. 

In addition, it would be good if authors could also elaborate on the presence or absence of statistically significant differences among participants with different demographic characteristics, and how these findings are related to findings from other relevant studies.

Finally, a proofreading of the final manuscript would assist in eliminating some minor English syntax and style mistakes that have been spotted.

In a nutshell, the work is interesting and well-reported. Some minor modifications would further improve the manuscript and its uptake by the scientific community.

Author Response

In the document, we provide point-by-point responses on how we have addressed each comment from the reviewer. We sincerely hope that the changes in this revision meet the review panel’s expectations. Thank you again.

Reviewer 3 Report

The authors should explain how did they selected the 30 users of m-health apps to conduct the pilot test of the questionnaire. Do they have statistic significance ? If so, please demonstrate.

The self-efficacy and response efficacy can be also related the the apps design and human interfaces. Please explain if you considered this point. 

Additional Comments:

1- The authors should demonstrate that the 30 cases of the survey pilot cohort have statistical significance and are representative of all sample. Otherwise it is not possible to assume the validity and reliability of the questionnaire. lines [367-374] of document.  

2- Regarding the self-efficacy and response efficacy related with hypothesis H1 [217-218]  and H2 [219-220] and  the  interface ergonomy of the apps used for the research,  this dimension was not evaluated. Definitely the usability of the interfaces may have an influence on the end-users self assessment. So I think that the authors should consider this variable too.

Author Response

In the document , we provide point-by-point responses on how we have addressed each comment from the reviewer. We sincerely hope that the changes in this revision meet the review panel’s expectations. Thank you again.

Reviewer 4 Report

The paper proposes a framework to evaluate continued intention of using m-health apps with regard to protection motivation theory and network externalities.
The proposed approach is interesting and the topic is relevant for the community.
Several hypothesis were done built on literature review as well as the questionare itself. A significant amount of people were interviewed and their taxonomy explained.
Finally, results are properly discussed. 

However, the paper has some drawbacks to be addressed prior to its publication:

- lines 72-84, page 2: it should be a numbered list, not a sequence. Please check the template.
- line 174, page 4: SNS acronym is not defined, please fix it. 
- line 190, page 4: Figure 1 must be described both in the text and in the caption. What are the blocks? what do dashed lines mean? what do the arrows stand for? It is important to properly explain it before you start analyzing its subblock in Sec. 3 subsections. 
- lines 211-214, page 5: very long sentence, hard to be read. Please rephrase.
- lines 217-220, page 5: hypothesis must be put in a bullet list. the same holds for the other hypotheses in the remaining of the paper. 
- lines 331-338, page 8: why H8 hypotheses do not appear in Fig. 1? For sake of clarity, and to give them the same importance of the others, you should add them in the figures (Fig. 1 and 2).
- line 354, page 8: use lowercase for Behavioral and Attitude, as you've done per the remaining items of the construct list. 
- line 366, page 9: who are the two professors? Are two paper authors? otherwise, they should be put in the acknowledgment section.
- lines 384-386, page 9: very long sentence, hard to be read. Please rephrase.
- page 11, Fig. 2: table cannot be part of the figure, it should be a separate table, properly referred and commented in the paper.
- page 11, table 4: column labels cannot be randomly split. Use new lines to avoid it; as a side effect, table width will shrink to a more suitable size.  
- Study limits and future works should be properly addressed into a subsection of Sec. 5.

Minor concerns:
- choose a convention, either m-health or M-health and apply it all over the paper; personally, I suggest you to use the former option. 
- do not get confused among the use of e.g. and i.e.: the former means "for example", the latter "that is".
- please, carefully check punctuation (e.g. lines 216, page 5, you miss a colon), new lines (e.g. table 4 column labels)  and upper/lowercase (e.g. line 184, page 4). do not forget the use of semicolons.

Finally, English must be deeply revised by a professional service or native speaker. Current level is quite far to be acceptable. 

Author Response

(The authors gave the same response as above.)

Round 2

Reviewer 3 Report

Maybe some questions about the profile of each person will improve the results and final conclusions.